# Generative Adversarial Networks Can Create High Quality Artificial Prostate Cancer Magnetic Resonance Images

**DOI:** 10.3390/jpm13030547

**Published:** 2023-03-18

**Authors:** Isaac R. L. Xu, Derek J. Van Booven, Sankalp Goberdhan, Adrian Breto, Joao Porto, Mohammad Alhusseini, Ahmad Algohary, Radka Stoyanova, Sanoj Punnen, Anton Mahne, Himanshu Arora

**Affiliations:** 1John P Hussman Institute for Human Genomics, Miller School of Medicine, University of Miami, Miami, FL 33136, USA; 2College of Medicine, University of Central Florida, Orlando, FL 32816, USA; 3Department of Radiation Oncology, Miller School of Medicine, University of Miami, Miami, FL 33136, USA; 4Department of Urology, Miller School of Medicine, University of Miami, Miami, FL 33136, USA; 5The Interdisciplinary Stem Cell Institute, Miller School of Medicine, University of Miami, Miami, FL 33136, USA

**Keywords:** generative adversarial networks, machine learning, MRI, image segmentation

## Abstract

The recent integration of open-source data with machine learning models, especially in the medical field, has opened new doors to studying disease progression and/or regression. However, the ability to use medical data for machine learning approaches is limited by the specificity of data for a particular medical condition. In this context, the most recent technologies, like generative adversarial networks (GANs), are being looked upon as a potential way to generate high-quality synthetic data that preserve the clinical variability of a condition. However, despite some success, GAN model usage remains largely minimal when depicting the heterogeneity of a disease such as prostate cancer. Previous studies from our group members have focused on automating the quantitative multi-parametric magnetic resonance imaging (mpMRI) using habitat risk scoring (HRS) maps on the prostate cancer patients in the BLaStM trial. In the current study, we aimed to use the images from the BLaStM trial and other sources to train the GAN models, generate synthetic images, and validate their quality. In this context, we used T2-weighted prostate MRI images as training data for Single Natural Image GANs (SinGANs) to make a generative model. A deep learning semantic segmentation pipeline trained the model to segment the prostate boundary on 2D MRI slices. Synthetic images with a high-level segmentation boundary of the prostate were filtered and used in the quality control assessment by participating scientists with varying degrees of experience (more than ten years, one year, or no experience) to work with MRI images. Results showed that the most experienced participating group correctly identified conventional vs. synthetic images with 67% accuracy, the group with one year of experience correctly identified the images with 58% accuracy, and the group with no prior experience reached 50% accuracy. Nearly half (47%) of the synthetic images were mistakenly evaluated as conventional. Interestingly, in a blinded quality assessment, a board-certified radiologist did not significantly differentiate between conventional and synthetic images in the context of the mean quality of synthetic and conventional images. Furthermore, to validate the usability of the generated synthetic images from prostate cancer MRIs, we subjected these to anomaly detection along with the original images. Importantly, the success rate of anomaly detection for quality control-approved synthetic data in phase one corresponded to that of the conventional images. In sum, this study shows promise that high-quality synthetic images from MRIs can be generated using GANs. Such an AI model may contribute significantly to various clinical applications which involve supervised machine-learning approaches.

## 1. Introduction

Prostate cancer is the most common cancer among men and the second most common cause of cancer-related death for men in the US [1]. A relatively high incidence of prostate cancer has invoked discussion about a national screening program [2]. The National Cancer Institute alone was estimated in 2020 to have spent $209.4 million on prostate cancer research and $233.2 million on clinical trials [3]. Despite this, the costs of treating prostate cancer are increasing more rapidly than for any other cancer [4]. Multiple studies have suggested that the high cost and time investment of prostate cancer clinical trials pose a significant barrier to research, as well as hindering the formation of large cohorts of patients with multiple years of follow-up that facilitate the drawing of high-value conclusions [5].

The use of Magnetic Resonance Imaging (MRI) in prostate cancer research and treatment is effective for prognosis, diagnosis, and active surveillance, and contributes to a reduced need for biopsy procedures in lower-risk patients [6]. Additional benefits of MRI usage include ease of visualization, staging, tumor localization, risk stratification, active surveillance monitoring, and detection of local failure after radiation therapy [7]. Prostate MRIs provide clearer and more detailed images of soft-tissue structures of the prostate gland than other imaging methods, making them valuable data for the research [8].

The Prostate Imaging Reporting and Data System (PI-RADS) is an established image-based scoring system that scores the probability of clinically significant prostate cancer on MRIs to guide the management of the disease [9]. Image fusion technology allows the combination of the high-quality soft tissue contrast resolution of MRIs with real-time anatomical depictions using computed tomography or ultrasounds [10,11]. This allows the precise mapping of PCa for biopsy and treatment. Artificial intelligence (AI) provides numerous opportunities for automating the lesion depiction that could increase the reproducibility of the PI-RADS and enhance diagnostic and treatment methods [12]. In this context, we have also developed an automated, quantitative mpMRI-based method to objectively guide dose escalation for high-risk habitat volumes based on prostatectomy GS [13]. Like other machine learning models, our mpMRI-based models were also trained with clinical data. There are a few limitations to relying on clinical data for training the models, including the time it takes to capture the data, the specificity of data with respect to the question being focused on, and the cost associated with conducting trials [14]. The solutions to these questions are being sought through the use of advanced machine learning models.

When applied to diagnostic imaging, AI has shown high accuracy in prostate lesion detection and prediction of patient outcomes such as survival and treatment response [15]. Machine learning can help medical experts in early disease detection; for prostate cancer, for example, machine learning classifiers predicted Gleason pattern 4 with approximately the same, if not more, accuracy than experienced radiologists [16]. Other deep learning approaches, such as convolutional neural networks (CNNs) and generative adversarial networks (GANs), have been used in various applications of medical image synthesis of PET, CT, MRI, ultrasound, and X-ray imaging, particularly in the brain, abdomen, and chest [17]. A recent study using synthetic MRI in colorectal cancer also found no difference in signal-to-noise ratio, contrast-to-noise ratio, and overall image quality between conventional and synthetic T2-weighted images [18]. GANs have also been shown to create realistic synthetic brain MRIs [19]. All these facts combined contribute to our hypothesis that we can use deep learning image synthesis to create medical images that translate into clinical use while avoiding the tremendous costs associated with repeat follow-ups. Synthetic prostate MRI using GANs is a possibility for large-scale machine learning projects, where millions of image samples can be generated without the need for millions of patients.

Therefore, the present study is the first step towards demonstrating that GANs have the potential to create synthetic images capable of simulating conventional prostate MRIs. With further development, GANSs will also decrease the number of patient follow-ups, reduce the costs associated with conducting/capturing data, and assist the radiologists in making decisions.

## 2. Materials and Methods

### 2.1. Prostate MRI Images

Prostate MRI images used in this study were obtained from three sources: (1) T2W tse (T2 weighted Turbo-Spin-Echo) images downloaded from the first 60 patients of the Cancer Imaging Archive’s ProstateX challenge repository [20]; (2) T2W fse (T2 weighted Fast-Spin-Echo Transversal) images, including manually segmented contour data from 45 patients enrolled in the fusion database from the University of Miami’s Sylvester Cancer Center; and (3) T2W fse images from 79 patients enrolled in the fusion database from the University of Miami Department of Urology. All images used were displayed in the axial plane. All human investigations were carried out after the IRB approval by a local Human Investigations Committee and in accordance with an assurance filed with and approved by the Department of Health and Human Services. Data have been anonymized to protect the privacy of the participants. Investigators obtained informed consent from each participant.

### 2.2. Image Preprocessing

Digital Imaging and Communications in Medical Science (Dicom) information for prostate MRIs from each dataset was converted into a nearly raw raster data (NRRD) format for better data processing, normalized to intensity values from 0–1, and then saved into 2D images from slices of the prostate along the z-axis using Python’s Matplotlib [21] package. To control MRIs of different dimensions and pixel sizing, images were cropped proportionally to the smallest dimension (x or y) into a square and then resized to a 500 × 500-pixel image using Python’s Pillow package.

### 2.3. Synthetic Image Generation

For each prostate, the 2D image used to train a generative model was the slice from the z-axis index with the largest manually identified prostate. Synthetic images were created using Generative Adversarial Networks for each preprocessed image with SinGAN [22] [https://github.com/tamarott/SinGAN] (accessed on 10 January 2022) under default settings. Using the 500 × 500 images in SinGAN would result in 10 generative models for each image. The random sample synthetic images produced in the 8th, 9th, and 10th editions of the models were considered for further analysis.

SinGAN created synthetic images that maintained the resolution of conventional MRIs. The images created used generative adversarial networks, invented by Goodfellow et al. [23]. SinGAN takes advantage of two neural networks: a Generator to generate image samples and a Discriminator to discriminate between conventional and synthetic samples. The training was performed on one image in a course-to-fine manner to make a model that could generate random samples (Figure 1).

We used a dataset of 790 conventional prostate MRIs and employed SinGAN to generate 237 synthetic images. From these images, 592 conventional and 178 synthetic images were selected for the training dataset. Additionally, a random selection of 25% from both the conventional and synthetic images, resulting in 198 conventional and 59 synthetic images, was used for testing purposes. Synthetic images appear to be topologically similar to the training image but have noticeable realistic variation, especially in the prostate and peripheral zone (PZ). Prostate and peripheral zone image and radiomic statistics are commonly used in studies involving prostate cancer risk, progression, tumor analysis, and clinical trial studies using radiation therapies [24]. This is due to prostate cancer appearing as a hypo-intense signal compared to a higher signal depicting a normal prostatic tissue [25].

### 2.4. Deep Learning Image Segmentation

Deep learning image segmentation of prostate MRI has been used in past years with relative success [26]. We hypothesized that a neural network could successfully segment high-quality synthetic images. Therefore, an image segmentation neural network was trained on the prostate contours of 45 patients from the University of Miami’s Sylvester Cancer Center dataset. For each patient’s prostate MRI, the three middle slices of the contour and corresponding T2 image were used to train the network, totaling 135 images in the training set. The neural network was trained using the PyTorch [27] machine learning package, and the code was adapted from the existing code [28]. The learning rate for each step of the model’s gradient descent was 0.00001, and the batch size was 3.

The trained model subjected the synthetic images to a quality control check. For the synthetic image to pass the deep learning image segmentation quality check, the output prediction boundary for the prostate had to have been at least 10,000 pixels and be a single, unbroken boundary. The rationale for 10,000 pixels comes from our training dataset having a minimum contour pixel size of 10,932.

### 2.5. Quality Control Study

A blind quality control test was given to selected participants with varying expertise in prostate cancer MRI research to test the realism of the synthetic images. The first group consisted of two scientists considered experts in the field who had 9 and 13 years of experience with prostate cancer MRIs. The second group consisted of 2 scientists who each had approximately one year research experience in prostate cancer MRIs, which we classified as having some experience but not yet expertise. Negative control was given to a third group of 4 scientists with no experience working with prostate cancer MRIs. The first quality control test was used on synthetic images manually selected from the output of the trained models from the 79 images obtained from the Urology department training set. In this first test, high-quality synthetic images were manually selected, while synthetic images with high distortion were excluded. A total of 60 images were used for this test, including 25 conventional and 35 synthetic images. The radiation oncology team members were given five seconds to study each image before they recorded their prediction as conventional or synthetic.

To improve the fidelity of the study, a second quality control test was given to the same participants to test for variance and to see if they would have learned to better differentiate between conventional and synthetic images with more exposure. The images used this time were subjected to the deep learning image segmentation pipeline and then randomly selected without replacement for inclusion to both automate the process and control for human bias in image selection. Synthetic images that were generated from images and passed the neural network criterion but were not in the training data were pooled and randomly selected for this test. The test consisted of 26 conventional images and 36 synthetic images, and participants were given the same format as the first assessment.

## 3. Results

### 3.1. Deep Learning Image Segmentation

To create a pipeline that automates a process for determining if a synthetic image was high enough quality to be included for grading by fellow scientists and in future projects, as well as to test whether synthetic prostate MRIs can be used in similar machine learning contexts as conventional prostate MRIs, the image segmentation neural network model was trained on 135 images from 45 patients for 3000 steps. For each step of the training, a T2W image and corresponding contour were given as inputs (Figure 2A). The contour is a binary file of the same dimensions as the MRI. After 15 epochs of training, the neural network predicted the contour with a Dice similarity coefficient ($DSC=\frac{2TP}{\left(2TP+FP+FN\right)}$) of 0.8857 and accuracy ($\frac{TP+TN}{TP+TN+FP+FN}$) of 0.9795 (Figure 2B). The 1000th step was predicted with a DSC of 0.9574 and an accuracy of 0.9945. The 3000th step was predicted with a DSC of 0.9991 and an accuracy of 0.9988. Although the model over-fit the training data, we still saw an increase in the prediction accuracy on synthetic images with more epochs of training and decided to use the 3000th step model for this analysis. For a synthetic prostate MRI image to pass the neural network segmentation pipeline’s criteria, the predicted segmentation had to have 10,000 pixels (4%) and one unbroken contour (Figure 2C). The 10,000-pixel cutoff was based on the training data contours’ minimum 10,932 pixels (4.3%) and median 19,481 pixels (7.8%). The segmentation pipeline performed well in not predicting segmentation in synthetic images which subjectively had obscured and malformed prostates. Out of 654 synthetically generated images, close to 39% (253, or 38.6%) passed the neural network pipeline’s criteria. The images that did pass the criteria were used in the further analysis of this study (Figure 2D).

### 3.2. CNN Quality Control and Applicability Validation

Next, to delineate the usability of the synthetic images from prostate cancer MRIs, we used a simple convolutional neural network. The first query the CNN was subjected to was to differentiate between conventional and synthetic images. For this, the CNN reached an accuracy of 76.4% in differentiating between the two image types. The second query the CNN was subjected to was to differentiate between tumor and normal images. For this, the CNN reached an accuracy of 77.67% in differentiating between the two image types. Together, no real difference was detected with regard to the sample source. The third query the CNN was subjected to was to evaluate if the synthetically generated images were of good enough quality. For this, the images (synthetic and conventional) were subjected to anomaly detection, where the anomaly represents a cancerous growth. From within the real images, the CNN was 84.37% accurate in detecting tumor images. Similarly, the CNN was 88.86% accurate in detecting tumor images from within the real images. Importantly, even though we used phase one, quality control-approved synthetic images, the rate of anomaly detection of CNN on synthetic images corresponded to that of the conventional images (Figure 3).

### 3.3. Round 1 Quality Control Test

The first-round quality control test was given to selected participants based on their experience with prostate cancer MRIs. Two participants had roughly a decade of experience (nine years, thirteen years), two participants had about a year of experience, and four participants had no experience. Participants were shown 60 hand-picked images, both synthetic and conventional, and asked after 5 s to give their opinion if the image was synthetic or conventional. Results from this experiment are summarized in Table 1. The accuracy for groups were 62%, 55%, and 53%, respectively. A Pearson’s Chi-square test showed no significant association between experience level and the number of correct predictions (*p* = 0.2855), no association between experience level and the number of false negatives (*p* = 0.4125), and no association between experience level and the number of false positives (*p* = 0.6675); however, a significant association was found for concordance within groups when all groups were considered. No significant association in concordance was determined when comparing the participants with ten years of experience to participants with one year of experience. These results demonstrate that for all levels, correctly identifying the difference between synthetic and conventional prostate MRI images is challenging.

### 3.4. Round 2 Quality Control Test

Figure 4 shows the workflow of this second quality control test. The second quality control test aimed to eliminate human bias in image selection (selecting the most realistic synthetic images and the lower quality conventional images), to automate the process, and to test if the participants had learned the difference since participating in the first test. The second quality control test was given to the same team of scientists. Out of 654 generated synthetic images, 253 met the criteria from the deep learning image segmentation pipeline and were randomly selected for the test. Table 1 illustrates the results of our quality control tests. Our three groups of participants scored correctly 67%, 58%, and 50%, respectively. A Wilcoxon signed ranked test shows that the number of correct scores between participants’ first and second surveys is not statistically different (*p* = 0.725), regardless of group. A Pearson’s Chi-Square test shows that there is an association between experience and group when considering all three groups (*p* = 0.005682); however, there was no significant association between those with ten years of experience and those with one year of experience (*p* = 0.1824). Furthermore, the group with no experience studying prostate cancer MRIs had high variance between scores, with the highest score matching the average of the expert group (67%), yet the lowest score was 35%. There was no significant association between false negatives (*p* = 0.2766) or false positives between experience levels (*p* = 0.06051). These results mirror a similar study where expert and basic prostate radiologists found no difference in diagnostic performance between conventional MRI and synthetic MRI [29]. They also show a similar result to another study where synthetic brain MRIs were created using a different GAN called DCGAN [20]. There was an association between the data’s experience and the respondents’ concordance (*p* = 0.001). Between the first and second tests, the number of false positives decreased as expected; however, the number of false negatives increased, where 47% of all synthetic images were incorrectly reported as true, with 40% of false negatives in the expert group. These results further support that synthetic prostate MRIs have realistic quality and that the process for selecting higher quality synthetic prostate MRIs can be automated.

### 3.5. Radiologist Quality Control Check

To further assess the quality of the synthetic images, we enlisted the help of a board-certified radiologist to grade our prostate MRIs based on quality. Thirty images (20 synthetic, 10 conventional) that met the criteria of the deep learning segmentation pipeline were randomly selected and given to the radiologist, who graded the images on a 1 to 10 scale based on the quality of being able to read and make a report (10 being best). The radiologist was NOT informed that the image set contained synthetic images. The mean grading was 6.2 for synthetic images and 5.5 for conventional images. A student t-test fails to reject the null hypothesis that the mean quality of synthetic and conventional images equals (*p* = 0.4839). The minimum, median, and maximum for synthetic and conventional images are (1, 6, 10) and (1, 5.5, 10), respectively. These results suggest that the synthetic images are of equal quality when blindly given to a radiologist. Figure 5 summarizes the pipeline of the corresponding generators and discriminators in the SinGAN network.

## 4. Discussion

Our studies show that current machine learning technologies such as Generative Adversarial Networks can be used to generate synthetic prostate cancer MRIs that mimic conventional data, which could have important implications for research and clinical practice in the field. Other machine learning applications, such as deep learning semantic segmentation, can be used to automate the process of filtering out higher-quality synthetic images. Due to the availability of sophisticated GAN models capable of performing lesion classification and tumor segmentation, the present study focused on the functional applications of GAN models and how they can improve the efficacy of digital imagery.

The present study is the first to compare the accuracy of synthetic and conventional MRIs of prostate cancer using different radiologists’ experiences. Moreover, we demonstrated that participants did not have better performance on the quality control assessment when synthetic images were manually selected compared to when the images were randomly pooled after passing the segmentation pipeline. This demonstrates the success of the pipeline because the human selection of the best synthetic images in the first visual assessment showed no difference in scores compared to the AI-segmented synthetic images in the second assessment. Additionally, the synthetic images that passed the deep segmentation pipeline showed no difference compared to conventional images when graded based on quality by a board-certified radiologist. Our results show promise for future studies that involve synthetic imaging.

The filtering process for high-quality synthetic images could be improved by using more training data in the segmentation neural network. Other studies involving automatic segmentation of the prostate with larger training databases show success in segmenting 2D and 3D prostate MRIs [30], as well as MRIs coming from multiple sources [26], such as our dataset. With only three slices each from 45 patients, the training data were relatively small and repetitive, which caused some visually realistic synthetic images to not meet the criteria. Furthermore, the criteria that the predicted prostate boundary had to be unbroken disqualified many otherwise realistic looking synthetic MRIs. Other studies that included larger training databases also have predicted segmentations that were not one single boundary, but were occasionally broken into pieces [31]. Since deep learning segmentation is not completely accurate, quality scores can be given prediction boundaries that may be a better criterion in the future than the criteria used in this preliminary study [31]. A segmentation model that considers other organs of the MRI, such as the bladder, urethra, gluteus maximus, rectum, or femur, can also be implemented to further select prostates with no distortion, although it will make the model highly selective and prone to overfitting. In turn, compressing the image more around the prostate can help reduce the amount of area the model needs to consider, and the amount of potential distortion, leading to enhanced segmentation. However, even with a very basic segmentation model, the model’s ability to segment the prostate proved an effective method in selecting high-quality synthetic images. This is shown in Table 1, where 47% of synthetic images were mistaken for conventional, compared to 40% by experts. These results show promise of refinement of the model in future studies, where more accurate detection of prostate segmentation and the surrounding organs will enhance the filtering of high-quality synthetic images.

In our study, the vendor used for the MRI and normalization programs was not stressed because we intended for participants to be presented with a wide variety of all types of MRI modalities that they may encounter. Furthermore, different versions of prostate MRI can be used, such as dynamic contrast-enhanced (DCE) perfusion imaging, where apparent diffusion coefficients (ADC) are commonly taken together [32] on top of the T2 diffusion-weighted imaging used in this study, or PET scans, commonly used in other studies. The premise of this study can be extrapolated to these different image modalities, which may yield better results due to the decreased complexity of ADC images compared to T2W. Furthermore, the use of GANs in medical imaging studies is expanding to include 3-dimensional capabilities [33], where our study was limited to 2-dimensional slices of prostate MRI. Furthermore, a survey with more scientists and radiologists would increase the power of these results. There was no statistical difference in the ratings of the quality of synthetic and conventional MRI by the board-certified radiologist, but we stressed the promising results of the prostate cancer researchers due to their specificity with the prostate cancer MRI in particular. A different questioning method could be used, similar to the quality of reading recording we asked the board-certified radiologist, or a Likert scale could be used as a scoring approach rather than binary responses [34]. Additionally, our study had participants study images for 5 s; a longer assessment time per image should be tested to see if accuracy increases significantly with more consideration. Lastly, the 60 images used in the quality control tests contain only a small proportion of the total generated synthetic images (34 out of 253), so participants did not see the full array of synthetic images that passed the neural network criterion.

Lesion classification and tumor segmentation are also newfound capabilities of AI, where models such as random forest, naïve bayes, linear support vector machine (SVM), and CNNs have been able to show Gleason grading accuracy comparable to pathologists [35]. These results support SinGAN’s unsupervised model using one training image, reducing tumor heterogeneity compared to models that combine multiple samples. Online databases such as those of the prostateX [20] challenge combine multiple views and modalities, such as the ADC image, Sagittal, and Coronal T2W images, on top of the T2W images used in this study. They also include lesion data with all levels of Gleason grading that can be potentially classified using the above solutions in synthetic images. Future machine learning studies will be conducted to test if high-quality synthetic images can be integrated and complement conventional images in classification, segmentation, and lesion detection.

## 5. Conclusions

We have demonstrated that synthetic prostate cancer MRI can be generated with high quality, mimicking conventional images. This process can be scaled to generate millions of unique synthetic samples. Other machine learning approaches, such as the deep learning segmentation model, can be used to remove synthetic images with high levels of distortion. This preliminary study suggests that synthetic prostate MRI images may be used in more complex imaging studies with clinical applications in the future.

## Figures and Tables

**Figure 1 jpm-13-00547-f001:**
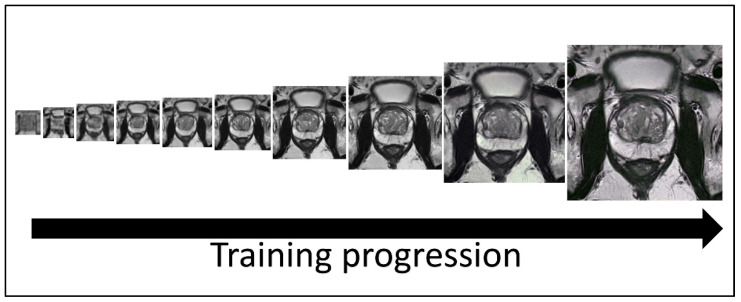
Image generation using SinGAN. SinGAN trains a model using generative adversarial networks in a course-to-fine manner. At each scale, the generator makes sample images that cannot be distinguished in down-sampled training images by the discriminator. The 8th, 9th, and 10th scale images (the three rightmost images) of the model were resized and used in all quality control tests.

**Figure 2 jpm-13-00547-f002:**
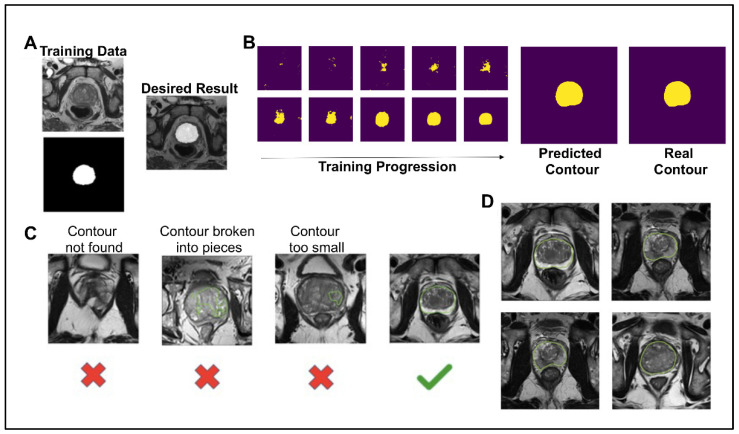
(**A**) Example of training data for the image segmentation neural network and the desired result. The top left is the T2 tse training image, and the bottom left is the corresponding binary contour. (**B**) The deep learning image segmentation training progression is shown for one image of the training set. The final predicted contour had a dice similarity coefficient of 0.99 to the real contour. (**C**) To pass the deep learning segmentation pipeline, the predicted contour had to be one continuous contour greater than 10,000 pixels. (**D**) Examples of the 253 synthetic images that passed the deep learning segmentation pipeline, shown with the prediction boundary.

**Figure 3 jpm-13-00547-f003:**
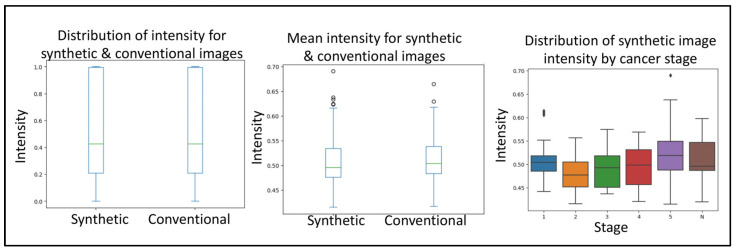
Boxplots depicting the minimum, median, interquartile range, maximum and potential outliers of image intensity for both synthetic and conventional samples. Highlighting the convolutional neural network quality control and applicability validation steps in evaluating the distribution of intensity for synthetic and conventional images, mean intensity for synthetic and conventional images, and distribution of synthetic image intensity by cancer stage. Here “N” represents normal and 1–5 represents the increasing progression of the disease.

**Figure 4 jpm-13-00547-f004:**
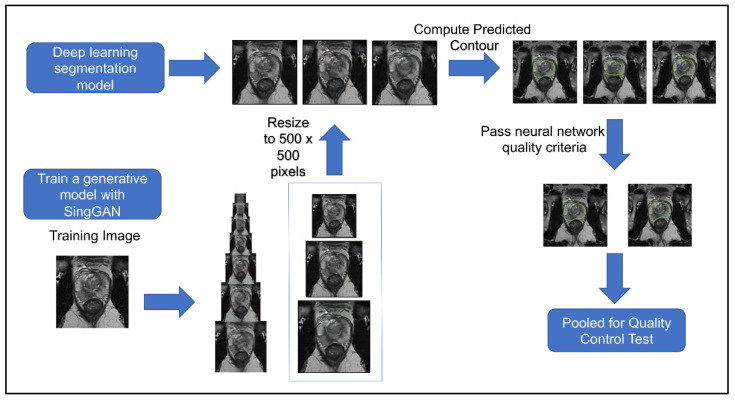
Workflow of procedure for second human quality control assessment. Three synthetic samples output by the last three generative models from SinGAN were resized to 500 × 500 and then given a predicted segment of the prostate using a pre-trained segmentation neural network. Synthetic images that had a segmentation of the contour that passed our defined criteria were pooled and randomly selected for our human visual assessment.

**Figure 5 jpm-13-00547-f005:**
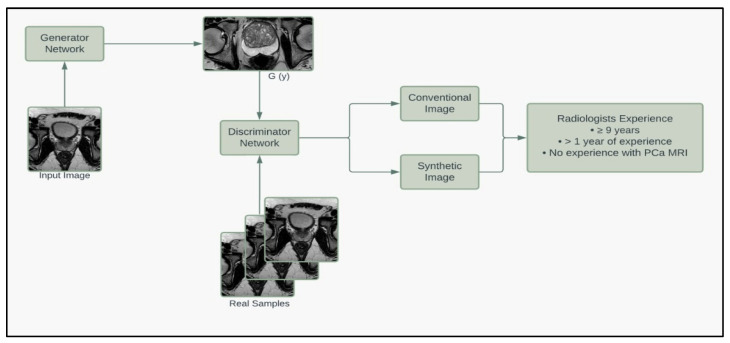
Model framework diagram of the corresponding generators and discriminators in the SinGAN network.

**Table 1 jpm-13-00547-t001:** The results from the first and second rounds of quality control tests. The second round used the deep learning segmentation pipeline and random sampling for the test creation. False positives are defined as conventional images that participants labeled as synthetic. False negatives are defined as synthetic images that participants labeled as conventional. Concordance between the participants is defined as the proportion of the same answers.

	Round 1			Round 2		
Amount Conventional	25			26		
Amount Synthetic	35			34		
Experience Level	10 Years	1 Year	No Experience	10 Years	1 Year	No Experience
% Correct	62	55	53	67	58	50
% FP	46	46	54	25	35	47
% FN	33	44	42	40	47	54
% Concordance	67	57	42	80	60	30

## Data Availability

Research data is stored in an institutional repository and will be shared upon request to the corresponding author.

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
