# Peer review of "Generative Adversarial Networks Can Create High Quality Artificial Prostate Cancer Magnetic Resonance Images"

_jpm, 2023, doi:10.3390/jpm13030547_

Round 1

Reviewer 1 Report (Previous Reviewer 3)

Good article.

Reviewer 2 Report (Previous Reviewer 2)

Accept in present form. 

This manuscript is a resubmission of an earlier submission. The following is a list of the peer review reports and author responses from that submission.

Round 1

Reviewer 1 Report

In this paper, the authors use GAN networks to synthesize new high-quality artificial prostate Cancer Magnetic Resonance Images, and the image data generated by the GAN network were initially filtered according to the segmentation results of the prostate boundaries in the synthesized MRI slices. Finally evaluate the quality of the synthesized MRI images by combining two rounds of manual empirical screening, and experimentally demonstrate the effectiveness of the method. In overall, the manuscript is easy to understand and the demonstrated results are relatively clear. However, this work is too simple, and the authors just simply use the existing GAN network model to enhance the MRI dataset, and the novelty of the core idea of the article is relatively low. There are some comments on this manuscript as follows.

 1.     The idea of using GAN networks for generating new medical datasets is not novel, and the authors can build on this paper to establish comparative experiments between conventional image datasets and enhanced datasets for applications such as lesion segmentation and recognition to verify the effectiveness of the method in enhancing the generalization ability of network models.

2.     The authors simply use the existing SinGAN to increase the number of MRI slices, on which the authors can build a GAN model more suitable for medical data augmentation and add comparison experiments.

3.      the model framework diagram of the corresponding generators and discriminators in the SinGAN network used should be added in section 3.2.

Reviewer 2 Report

The manuscript is already available in the following link: https://www.biorxiv.org/content/10.1101/2022.06.16.496437v1.full.pdf+html

Reviewer 3 Report

Personally I found this work very interesting and in line with recent innovations in terms of machine learning and artificial intelligence.

Reviewer 4 Report

the article does not have a surprising conclusion.

Lines 248 and 249 are related to the Submission guidelines and should be removed.

It is not standard to refer to others' articles in the conclusion section.

Reviewer 5 Report

The authors investigated to generate synthetic prostate cancer MRI images. This method can support other machine learning approaches in classification, segmentation and lesion detections of cancers. The developed method can generate synthetic images that support the training of deep learning models to segment the boundary of prostates as shown in the manuscript. The authors also tested the quality of the synthetic MRI images compared with conventional images with experienced, less experienced and no experienced scientists. The results indicates a relatively high quality synthetic MRI images generated using GAN. Overall, the manuscript is in high quality and the conclusions are supported by the results. The findings from the authors can be very useful to the machine learning studies with the efforts on synthetic MRI images. The manuscript is suggested to be accepted in the journal.